# Medication-Wide Association Study Plus (MWAS+): A Proof of Concept Study on Drug Repurposing

**DOI:** 10.3390/medsci10030048

**Published:** 2022-08-31

**Authors:** Yan Cheng, Edward Zamrini, Ali Ahmed, Wen-Chih Wu, Yijun Shao, Qing Zeng-Treitler

**Affiliations:** 1Department of Clinical Research and Leadership, George Washington University, Washington, DC 20037, USA; 2Center for Data Science and Outcome Research, Washington DC VA Medical Center, Washington, DC 20422, USA; 3Department of Neurology, University of Utah Hospital, Salt Lake City, UT 84132, USA; 4Division of Neurology, Irvine Clinical Research, Irvine, CA 92614, USA; 5Department of Medicine, Georgetown University, Washington, DC 20057, USA; 6Providence VA Medical Center, Providence, RI 02908, USA; 7Department of Medicine and Department of Epidemiology, Brown University, Providence, RI 02912, USA

**Keywords:** AD, ADRD, drug repurposing, hypothesis generation, hypothesis testing

## Abstract

The high cost and time for developing a new drug or repositioning a partially-developed drug has fueled interest in “repurposing” drugs. Drug repurposing is particularly of interest for Alzheimer’s disease (AD) or AD-related dementias (ADRD) because there are no unrestricted disease-modifying treatments for ADRD. We have designed and pilot tested a 3-Step Medication-Wide Association Study Plus (MWAS+) approach to rigorously accelerate the identification of drugs with a high potential to be repurposed for delaying and preventing AD/ADRD: Step 1 is a hypothesis-free exploration; Step 2 is mechanistic filtering; And Step 3 is hypothesis testing using observational data and prospective cohort design. Our results demonstrated the feasibility of the MWAS+ approach. The Step 1 analysis identified potential candidate drugs including atorvastatin and GLP1. The literature search in Step 2 found evidence supporting the mechanistic plausibility of the statin-ADRD association. Finally, Step 3 confirmed our hypothesis that statin may lower the risk of incident ADRD, which was statistically significant using a target trial design that emulated randomized controlled trials.

## 1. Introduction

The high cost and time for developing a new drug (~$3 billion and ~10 years) or repositioning a partially-developed drug (~$1 billion and ~5 years) [1] has fueled interest in “repurposing” drugs (from >20,000 approved drugs from about 3000 approved molecular entities) [2,3]. However, drug repurposing is often rare and accidental [4,5]. A purposeful, timely repurposing of an old drug for a new disease will likely benefit from a systematic and sophisticated examination of large comprehensive clinical databases with data on pharmacy fills and long follow-ups.

The need for drug repurposing is particularly pressing for Alzheimer’s disease (AD) or AD-related dementias (ADRD). The term ADRD is used here in the syndromic sense (progressive amnestic dementia for which there was no evidence for another etiology), and not biomarker supported [6,7]. Nearly 6 million Americans ≥65 years have ADRD [8,9]. There is no drug to cure AD/ADRD, and no new symptom-modifying drug was approved during 2003–2021 [10,11]. Recent (7 June 2021) FDA approval of aducanumab, based on a surrogate endpoint and using the Accelerated Approval Pathway reserved for serious or life-threatening illnesses, highlights the urgency of finding drugs for the prevention and treatment of AD/ADRD.

A number of prior studies have been carried out to repurpose existing drugs for ADRD [12,13]. Most of the studies have leveraged current knowledge of ADRD biology and the molecular mechanism of the drugs. These studies have utilized genomics, proteomics, metabolomics, and molecular biology in various ways seeking to identify promising candidates for repurposing. Clinical data was also utilized in a few studies.

Building on existing knowledge of mechanistic pathways may speed up the filtering of the thousands of approved drugs. However, the limitation in the case of ADRD is that researchers’ understanding of the mechanisms of drugs or pathology is far from complete. Relying on incomplete knowledge potentially hinders the discovery of new targets. It has been suggested that a more objective, “amyloid-agnostic” approach will be necessary to productively advance the field and target other mechanisms associated with neurodegeneration and AD/ADRD, such as viral infection, neuroinflammation, cerebral vascular dysfunction, diabetes, oxidative stress, and mitochondrial dysfunction [14,15].

We designed a 3-Step Medication-Wide Association Study Plus (MWAS+) approach to rigorously accelerate the identification of drugs with a high potential to be repurposed to delay and prevent AD/ADRD. Like genome-wide association studies (GWAS) [16], MWAS is a hypothesis-free process that links drugs to incident AD/ADRD. The hypotheses generated in the “MWAS” phase are then assessed for mechanistic plausibility and tested in the “+” phase with cutting-edge causal inference methods.

Step 1 is a hypothesis-free exploratory case-control MWAS to identify candidate drugs associated with AD/ADRD. Step 2 is mechanistic filtering, where domain experts examine the drug structures, targets, mechanisms of action, and relations to disease pathology to assess the mechanistic plausibility. Selected candidate drugs are then tested in hypothesis-driven studies in Step 3, applying design principles from randomized trials to analyze observational data.

In this proof-of-concept study, we chose the Veterans Affairs (VA) electronic health record (EHR) longitudinal data of >3 million Veterans ≥65 years (>200,000 African Americans; ~55,000 women) with ≥10 years of follow-up and ~200,000 incident AD/ADRD cases. These data contain ~600 prescription drugs (each used by ≥10,000 patients) with comprehensive pharmacy data with drug fill records. 

## 2. Materials and Methods

Study population: For Step 1, we began with a previously curated cohort of 4,045,269 Veterans, including 432,469 African Americans. The cohort was created based on Veteran Health Administration (VHA)’s Corporate Data Warehouse (CDW) data available at the VA Informatics and Computing Infrastructure (VINCI). We restricted our study sample to African and White Americans aged ≥65 years who regularly received care in the VHA healthcare system during 1999–2016 calendar years. Regular use of VHA service was defined as at least 1 hospitalization or 2 outpatient visits within 12 months before the index VHA encounter qualifying the Veteran’s entry into this cohort. 

The cohort also had the following inclusion criteria: (1) have at least one outpatient or inpatient encounter after attaining age ≥ 65 years between 1/1/1999 and 12/31/2016; (2) be free of diagnoses of AD/ADRD before the index encounter; and (3) be free of severe psychiatric conditions, namely, schizophrenia or bipolar disorder. A diagnosis of AD/ADRD was ascertained using International Classification of Diseases (ICD) codes (Appendix A). Veterans who had other dementia of known causes (ICD codes in Appendix A) were excluded. We further restricted the cohort to those with ≥10 years of history before their AD/ADRD diagnosis or study end encounter date, reducing the cohort to 3,295,711 patients. Amongst these, there are 199,368 AD/ADRD cases. Matching cases to controls at the ratio of 1 to 5 by age, sex, race, and ethnicity, we obtained a cohort of 1,196,226 patients for the Step 1 case-control study. 

Step 1: For exploration, the neurologist on the team selected 10 drugs that either have been discussed as potential treatments for AD/ADRD and/or are commonly prescribed to older adults. We first examined unadjusted associations of these drugs with cases and controls, by describing the prevalence of their use prior to ADRD diagnosis or last visit date. To examine if the associations of drugs with ADRD were incremental, we estimated cumulative doses (like pack-years of smoking) of the ten select drugs and then divided them into deciles. We then estimated odds ratios (ORs) for each decile using non-users as references. A *p* value of 0.005 (0.05/10 applying the Bonferroni correction for multiple testing) was considered significant. The following 10 medications met the *p* value cutoff threshold: Atorvastatin, Duloxetine, Finasteride, GLP1A, Losartan, SERM, Sildenafil, Terazosin, and Vitamin D3.

Step 2: One of the first classes of medication we chose to examine were the statins. Compared to individuals in the higher 5 deciles of cumulative exposure, atorvastatin cumulative use in the lower 5 deciles was associated with higher ORs of developing ADRD. We conducted a literature review of the mechanism of action of statins in Alzheimer’s disease. We also reviewed prior studies of statins’ effect on ADRD incidence and progression. 

Step 3: We tested the association between the initiation of statins (including atorvastatin) and incident AD/ADRD in a prospective cohort design. To attenuate selection bias, we began by identifying patients diagnosed with hyperlipemia. We used a Target Trial framework approach to assemble the study cohort to emulate the design of a randomized controlled trial using observational data so that causality can be attributed to the observed associations [17]. We “enrolled” patients into 4 simulated “trials” (annual between 2002–2005). In each trial, we “enrolled” all patients who were initiated on a statin and identified patients not initiated at a 1:1 ratio. The non-initiated patients’ index date was chosen as the date of a random prescription meeting the matching criteria. Subsequently, propensity scores for the initiation of statins were estimated in a logistic regression model using age, sex, race, ethnicity, marital status, income, living area, year of diagnosis, disease duration, comorbidities, body mass index, blood pressure, serum glucose, and lipid profile as covariates. We then used a matching algorithm to assemble a balanced matched cohort of 251,692 pairs (Figure 1). Finally, we used a Cox regression model to estimate the association of statin use and incident ADRD in the matched cohort.

## 3. Results

Step 1: The prevalence of use of most medications was lower in the ADRD cases compared to controls (Table 1). Figure 2 shows the associations of cumulative dose in deciles with incidence of AD/ADRD, relative to non-users of those drugs during the 10 years prior to the incident ADRD or the last visit. Atorvastatin (column 1, Figure 2), a statin used for high cholesterol, was associated with a modest 6% (OR of treated = 0.94) lower risk, but there was an incremental dose-response (based on cumulative exposure) in that association so that the magnitude of the associated lower risk was higher at higher doses and vice versa (Figure 2). In contrast, duloxetine (column 2, Figure 2), a selective serotonin and norepinephrine reuptake inhibitor used for depression, was associated with a 15% (OR of treated = 0.85) lower risk of ADRD overall, while there was no evidence of dose-response in that relationship.

Step 2: A literature review demonstrated significant interest in repurposing statins for ADRD. Mechanistically, statins have anti-inflammatory and antioxidant properties. Statins have been shown to reduce C-reactive protein (CRP) concentrations while elevated CRP are associated with dementia [18]. Statins are reported to have anti-oxidative effects by inhibiting the increase of 8-isoprostane and suppressing the activity of nitric oxide synthase. A possible path for antioxidants to reduce dementia risk is by reducing oxidative stress which is positively associated with impaired cognitive function [19]. It has also been suggested that statins may play a beneficial role in reducing amyloid β-induced neurotoxicity [20]. 

In observational studies, statins have also been linked to decreased risk of ADRD [21]. On the other hand, randomized controlled trials (RCT) have failed to find such beneficial effect [22]. The known limitations in observational studies and RCTs may explain the differences in the findings: Observational data analyses are prone to biases and confounding, and have traditionally been viewed as not capable to demonstrate causal relationships. RCT are typically regarded as gold standard, but often suffer from relatively limited sample sizes, not representative of the populations, and relatively short follow ups. In addition, different statins have somewhat different properties including blood brain barrier permeability and potency. Given the mixed evidence and continuing studies of statin, we consider the conflictive results on the association between atorvastatin and ADRD revealed in the Step 1 analysis plausible.

Step 3: The demographic and clinical characteristics of pre-matched and matched cohorts are shown in Table 2. Findings from our propensity score-matched cohort demonstrate that initiation of statin was associated with a slightly lower risk of incident ADRD (adjusted hazard ratio, 0.97; 95% confidence interval, 0.96–0.99; *p* = 0.0007) in the statin new initiator group. Shown in Table 3, the number of events is 13.0% in both the new statin group and the control group. The time to event is slightly shorter in the control group than new statin initiator group (8.5 vs. 8.8 years). Shown in Table 4 and Figure 3, the subgroup analysis suggests a bigger effect in other statins than simvastatin and lovastatin (i.e., atorvastatin, fluvastatin, pravastatin, rosuvastatin; adjusted hazard ratio, 0.89; 95% confidence interval, 0.83–0.95; *p* = 0.0009). 

Since most statins prescribed in the VHA from 2002–2005 were simvastatin, followed by lovastatin, the number of patients with other statin prescriptions including Atorvastatin was relatively small (*n* = 6980) due to the earlier approval dates of simvastatin and lovastatin compared to Atorvastatin in the US. Therefore, propensity score matching at the subgroup analyses was not performed. Instead, the subgroup analyses were adjusted for propensity scores in a Cox regression model. The findings confirm the observation from Step 1, which was corroborated by existing literature in Step 2 and confirmed with a simulated target trial design in step 3. 

## 4. Discussion

Significance: This pilot study demonstrated the feasibility of the MWAS+ approach. Our Step 1 analysis revealed drugs (i.e., atorvastatin and GLP1), the use of which can be associated with a lower risk of ADRD. However, it also identified drugs such as Terazosin (an alpha blocker) that have not previously been linked with ADRD risk. The literature search in Our Step 2 found a body of evidence for mechanistic plausibility of the statin-ADRD association. Finally, Step 3 confirmed our hypothesis that statin may lower the risk of ADRD, which was statistically significant, albeit a very small effect, using a target trial design that emulated randomized controlled trials.

The main focus of this paper is the MWAS+ approach, rather than the study of statins in ADRD. It is prohibitively costly to broadly test the large number of available drugs for conditions that the drugs were not approved for. In the era of big data, it is possible to selectively identify candidates for close examination. We developed MWAS+ as a generalizable approach to take advantage of the big data. Nevertheless, the step 1 analysis and step 2 literature review both revealed mixed findings on the statin-ADRD association. Dozens of observational studies have reported the reduction of 15 to 35% in the risk ADRD associated with statin use [23]. Clinical trials however did not show such effects [24]. The target trial design in Aim 3 is intended to mimic a randomized controlled trial and indeed produced results that were more aligned with the randomized controlled trial than with other observational studies. Our result from the target trial design is also consistent with a similar study that emulated a target trial of statin use and dementia risk using a dataset with a small number of initiators (*n* = 622).

Implication: Given the large number of drugs approved by the FDA and utilized in clinical practice, MWAS+ is likely to generate many hypotheses. Even though we will only be able to assess the mechanistic plausibility of the top candidates and further test the hypotheses of a small number of drugs with plausible mechanisms, our approach will open the door for future studies by other researchers. In addition, MWAS+ can be applied to other diseases and conditions that do not have effective treatments.

Limitation: Even though drug repurposing is a quicker process than drug development, drug candidates identified by MWAS will likely need to go through rigorous clinical trials. Furthermore, additional work will likely be needed to examine the risks and benefits, in the context of prevention vs. treatment group. 

As a proof-of-concept study, we conducted a brief literature review in Step 2 and only examined 1 drug in steps 2 and 3. Given the proof-of-concept nature, we did not explore promising drugs such as GLP1 in this project. GLP1 is currently being studied in clinical trial, however, its use as an ADRD treatment has not yet been proven.

Our dataset has a small percentage of women. Observational data analysis has been viewed by many as only able to identify associations due to the potential risk of residual confounding although in this case our target trial approach results approximate those of previous RCTs and could be useful for the study of future drug candidates which have limited RCT evidence. Using a target trial framework and target trial design can potentially allow us to make inferences on causal relationships.

Moreover, some prior literature suggested that sustained statin use is associated with reduced ADRD risks more than the initiation or intermittent use [25,26]. We chose to use the new user design because statins are typically used on a long-term basis. In addition, because statin prescription is common in patients with hyperlipidemia and most patients would receive a statin, an on-protocol design would result in a much shorter follow up in the control group than in the treatment. 

Future work: In step 3, we used a new user design, which does not consider the cumulative exposure. On the other hand, the heat map in step 2 suggests that a higher decile of cumulative exposure may be needed to affect incident ADRD. In future studies, we plan to use marginal structural modeling to count for the difference in cumulative exposure.

We also plan to enhance the MWAS+ in step 1 with deep learning and conduct a systematic and comprehensive review of the candidates. We would like to compare different statins and cumulative exposure levels in future studies regarding statins.

## Figures and Tables

**Figure 1 medsci-10-00048-f001:**
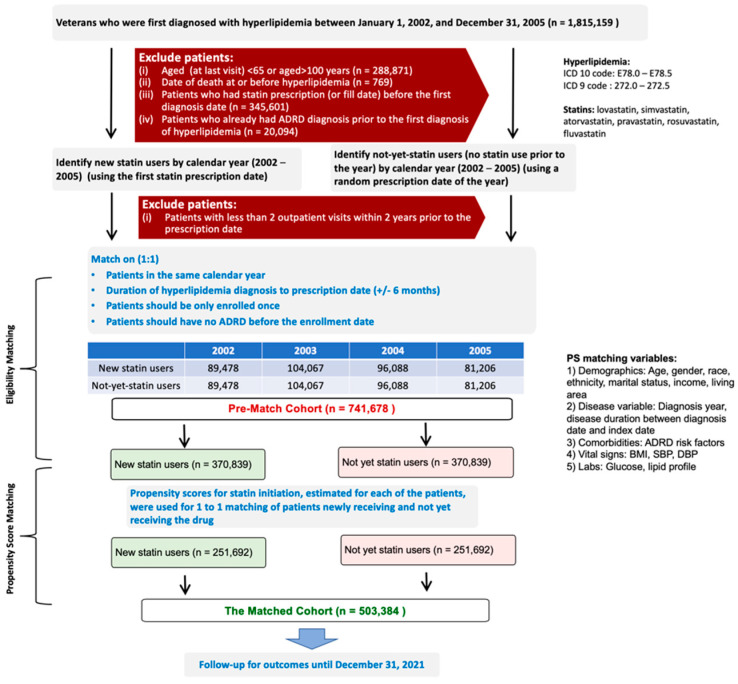
Cohort assembly.

**Figure 2 medsci-10-00048-f002:**
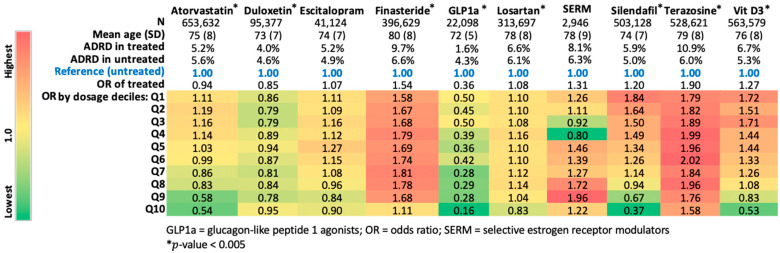
Heatmap of medication dose and ADRD risk. Of note, these data displayed are only to demonstrate the feasibility of MWAS and these are not adjusted associations.

**Figure 3 medsci-10-00048-f003:**
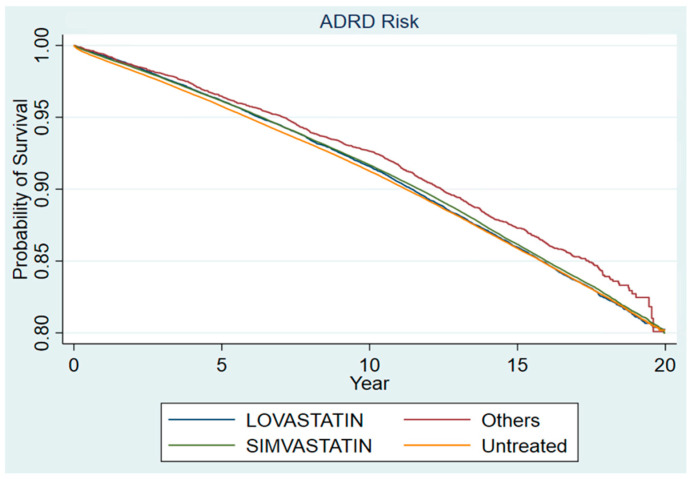
Kaplan-Meier survival estimates for statin and non-statin initiators.

**Table 1 medsci-10-00048-t001:** Unadjusted association between sample drugs and AD/ADRD.

Medication	AD/ADRD (*n* = 199,386)	No AD/ADRD (*n* = 996,840)
Atorvastatin	17.2%	18.8%
Duloxetine	1.9%	2.3%
GLP1a agonist	0.18%	0.53%
Gabapentin	21.5%	17.4%
Ibuprofen	21.2%	13.3%
Lisinopril	46.4%	33.1%
Losartan	10.3%	10.0%
Omeprazole	42.7%	31.6%
Sildenafil	15.0%	13.9%
Terazosin	28.7%	17.6%

**Table 2 medsci-10-00048-t002:** Demographic and clinical characteristics of the statin user cohort.

*n* (%) or Mean (±SD)	Pre-Match				Post-Match			
	No Statin *n* = 370,839	Statin *n* = 370,839	*p* Value	ASD	No Statin *n* = 251,692	Statin *n* = 251,692	*p* Value	ASD
Age	65.1 (±10.4)	66.6 (±9.7)	<0.001	15	65.9 (±10.3)	65.8 (±9.7)	<0.001	1
Female	13,708 (3.7%)	9344 (2.5%)	<0.001	7	7470 (3%)	7875 (3.1%)	<0.001	1
Race (African American)	36,926 (10%)	27,617 (7.4%)		9	21,208 (8.4%)	21,776 (8.7%)		1
Hyperlipidemia Duration	4.6 (±5.0)	2.6 (±5.2)	<0.001	39	3.8 (±4.4)	3.4 (±6.0)	<0.001	8
Unmarried	144,054 (38.8%)	125,892 (33.9%)	<0.001	10	90,052 (35.8%)	91,954 (36.5%)	<0.001	1
Rural residence	80,729 (21.8%)	77,751 (21%)		2	54,023 (21.5%)	54,268 (21.6%)		0
Income (1st Quartile)	90,856 (24.5%)	84,718 (22.8%)	<0.001	4	59,291 (23.6%)	59,801 (23.8%)	0.131	0
Income (2nd Quartile)	91,038 (24.5%)	88,826 (24%)		1	61,032 (24.2%)	61,234 (24.3%)		0
Income (3rd Quartile)	90,369 (24.4%)	91,592 (24.7%)		1	61,925 (24.6%)	61,725 (24.5%)		0
Income (4th Quartile)	92,107 (24.8%)	101,637 (27.4%)		6	66,150 (26.3%)	65,540 (26%)		1
Income (Unknown)	6469 (1.7%)	4066 (1.1%)		5	3294 (1.3%)	3392 (1.3%)		0
Alcohol abuse	30,960 (8.3%)	16,136 (4.4%)	<0.001	16	14,405 (5.7%)	14,592 (5.8%)	0.258	0
Smoking	77,224 (20.8%)	60,550 (16.3%)	<0.001	12	45,531 (18.1%)	46,455 (18.5%)	<0.001	1
Systolic Blood Pressure	137.7 (±15.7)	139.5 (±17.2)	<0.001	11	138.6 (±16.1)	138.5 (±16.6)	0.009	1
Diastolic Blood Pressure	77.0 (±9.4)	76.9 (±10.0)	<0.001	1	76.9 (±9.5)	77.0 (±9.7)	0.02	1
BMI	28.7 (±5.2)	28.8 (±5.0)	<0.001	2	28.8 (±5.2)	28.8 (±5.1)	0.955	0
Glucose, mg/dL	114.5 (±41.4)	117.6 (±44.5)	<0.001	7	115.9 (±42.8)	115.7 (±42.0)	0.228	0
Total cholesterol, mg/dL	200.4 (±39.1)	210.9 (±46.6)	<0.001	24	205.6 (±39.4)	205.2 (±45.0)	<0.001	1
LDL cholesterol, mg/dL	123.2 (±34.1)	135.1 (±39.8)	<0.001	32	129.2 (±33.9)	128.8 (±38.3)	<0.001	1
Triglycerides, mg/dL	188.0 (±157.3)	176.3 (±142.0)	<0.001	8	180.3 (±142.7)	179.7 (±154.7)	0.248	0
Hypertension	250,163 (67.5%)	251,338 (67.8%)	0.004	1	170,824 (67.9%)	170,495 (67.7%)	0.321	0
Ischemic heart disease	71,919 (19.4%)	114,524 (30.9%)	<0.001	27	60,321 (24%)	55,543 (22.1%)	<0.001	5
Atrial fibrillation	20,749 (5.6%)	17,912 (4.8%)	<0.001	4	13,380 (5.3%)	13,583 (5.4%)	0.204	0
Heart failure	17,430 (4.7%)	15,937 (4.3%)	<0.001	2	11,380 (4.5%)	11,342 (4.5%)	0.796	0
Diabetes	91,581 (24.7%)	106,616 (28.7%)	<0.001	9	67,568 (26.8%)	66,626 (26.5%)	0.003	1
Stroke	13,386 (3.6%)	16,579 (4.5%)	<0.001	5	10,054 (4%)	9895 (3.9%)	0.251	1
Chronic kidney disease	8367 (2.3%)	6174 (1.7%)	<0.001	4	4859 (1.9%)	4918 (2%)	0.547	1
Anemia	29,602 (8%)	16,915 (4.6%)	<0.001	14	14,738 (5.9%)	14,862 (5.9%)	0.458	0
Arthritis	109,390 (29.5%)	78,411 (21.1%)	<0.001	19	61,952 (24.6%)	64,391 (25.6%)	<0.001	2
Cancer	58,860 (15.9%)	42,834 (11.6%)	<0.001	13	33,865 (13.5%)	34,902 (13.9%)	<0.001	1
COPD	47,276 (12.7%)	35,662 (9.6%)	<0.001	10	27,367 (10.9%)	28,174 (11.2%)	<0.001	1
Depression	74,229 (20%)	49,195 (13.3%)	<0.001	18	39,780 (15.8%)	41,655 (16.5%)	<0.001	2
TBI	1251 (0.3%)	667 (0.2%)	<0.001	2	607 (0.2%)	587 (0.2%)	0.562	0

**Table 3 medsci-10-00048-t003:** Number of events, length of follow-up, time to event and incidence per 1000 patient-years in the non-statin vs. statin initiator groups.

Category	Number of Events (%)	Follow Up, Years, Mean (STD)	Time to Event among Patients Developing ADRD	Incidence, per 1000 Patient-Years
No Statins (*n* = 251,692)	32,699 (13.0%)	12.7 (5.6)	8.5 (5.2)	10.3
Statins (*n* = 251,692)	32,890 (13.0%)	13.0 (5.6)	8.8 (5.1)	10.0

**Table 4 medsci-10-00048-t004:** Hazard ratio estimates for statin initiation.

HR	Intention-to-Treat Unadjusted Analyses	Intention-to-Treat Adjusted for Propensity Score
Statins vs. no Statins	0.98 (0.96–0.99); *p* = 0.0020	0.97 (0.96–0.99); *p* = 0.0007
Simvastatin (*n* = 212,469) vs. no Statins	0.98 (0.96–0.99); *p* = 0.0039	0.97 (0.96–0.99); *p* = 0.0020
Lovastatin (*n* = 32,243) vs. no Statins	0.99 (0.96–1.02); *p* = 0.5716	0.99 (0.96–1.02); *p* = 0.4625
Other statins (*n* = 6980) vs. no Statins	0.90 (0.83–0.95); *p* = 0.0010	0.89 (0.83–0.95); *p* = 0.0009

## Data Availability

The datasets generated during and/or analyzed during the current study are not publicly available to protect the privacy of research participants but aggregated datasets are available from the corresponding author on reasonable requests.

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
