# Peer review of "Medication-Wide Association Study Plus (MWAS+): A Proof of Concept Study on Drug Repurposing"

_medsci, 2022, doi:10.3390/medsci10030048_

Round 1

Reviewer 1 Report

Cheng et al have described their study on repurposing of Alzheimer’s disease and related dementias. The manuscript described the study, method, data and their findings in a very methodical way. The literature has been thoroughly reviewed and made comprehensive by arranging in table form. More content can be added in the table legend if possible.  

The readers would benefit from addition of more discussion with examples in relation to their findings. Authors also need to address the limitations of their approach and also about the outcome drugs if repurposed. 

Author Response

Dear Reviewer,

Thank you for your feedback. We have strengthened the limitations section and discussed the outcomes as requested. The table legends were also revised to be more informative.

Reviewer 2 Report

This manuscript presents “proof of concept” results based on an approach termed “A 3-Step Medication-Wide Association Study Plus (MWAS+)” to accelerate the identification of existing drugs that can be repurposed for the treatment for Alzheimer’s disease (AD) or AD-related dementias. 

 The manuscript highlights the benefit of the MWAS+ approach as a proof of concept.  The manuscript also highlights the implication of the results as well as the current limitation of the results.  The MWAS+ approach has the potential to identify current drugs that has the potential to be repurposed for other diseases based on future studies by the same or other researchers.

The manuscript is well written with only minor spelling/grammar corrections suggested.  Furthermore, although the researchers indicate the potential benefit of the MWAS+ approach using atorvastatin as drug to be repurposed for AD they also mention GLP1.  GLP1 is, however, not discussed in the manuscript.  Why is GLP1 not discussed.  It should be discussed as well as this is very valuable for the “proof of concept” for the MWAS+ approach.   

Author Response

Dear Reviewer,

We thank you for your feedback. We did an additional round of review and editing. We also explained that while GLP1 is a promising drug, we did not examine this drug in this proof of concept manuscript.